# RELIABLE DETECTION OF AUTISM SPECTRUM DISORDER IN CHILDREN USING CONFORMAL PREDICTION

## ABSTRACT

Autism Spectrum Disorder (ASD) is a neurological condition affecting communication and daily functioning. Early intervention can reduce challenges in learning and behavior, motivating the use of machine learning techniques for ASD detection. Although models with high accuracy and F1 scores may appear promising, they can be misleading in low-prevalence settings. By Bayes' theorem, low prevalence substantially reduces the positive predictive value (PPV), meaning that even models with strong traditional metrics may yield unreliable predictions in practice. False-positive ASD detections can lead to unnecessary psychological stress, including anxiety and depression, while false negatives may delay intervention, making treatment more difficult later. In this paper, we integrated conformal prediction into the classification pipeline. Unlike standard classifiers, conformal methods provide prediction sets that include the true label with a specified confidence level $(1 - \alpha)$, thereby reducing the risk of false predictions. Results show that conformal predictors occasionally left cases unpredicted, thereby abstaining in situations where reliability could not be guaranteed. Among the evaluated models, SVM achieved the best performance with 86% correct predictions and 14% abstentions, followed by Logistic Regression (84% correct, 16% abstentions). These results demonstrate that conformal prediction offers a more trustworthy approach for ASD screening.

## 1 INTRODUCTION

Autism Spectrum Disorder (ASD) is a developmental condition that arises from a combination of genetic and environmental factors. It often leads to difficulties in communication and social interaction. ASD diagnosis depends on behavioral traits, such as the presence of restrictive, repetitive behaviors (RRBs). Early diagnosis is critical, as timely intervention can significantly improve long-term outcomes in learning and behavior. Neural measures have been used as complementary to behavioral traits (Avlund et al., 2021). Technologies include structural approaches such as magnetic resonance imaging (MRI) and diffusion tensor imaging (DTI), as well as functional techniques like functional MRI (fMRI), electroencephalography (EEG), and functional near-infrared spectroscopy (fNIRS). Although neuroimaging techniques have been applied to ASD detection, they are costly and time-consuming, making them infeasible for ASD screening (Eslami et al., 2019). As a result, machine learning methods have increasingly been explored for ASD screening, since they offer faster and less expensive alternatives. Vakadkar et al. (2021) applied machine learning algorithms, while Raj & Masood (2020) used deep learning approaches such as convolutional neural networks (CNNs) and achieved outstanding results. However, due to small sample sizes and the low prevalence of ASD (approximately 1% worldwide; (Zeidan et al., 2022)), these models suffer from a reduced positive predictive value (PPV). This means that false positives and false negatives remain a serious concern.

To address this limitation, we incorporate conformal prediction into the classification pipeline. Conformal prediction, originally introduced by Vovk et al. (2005), provides prediction sets that include the true label with a user-specified confidence level, thereby reducing the risk of unreliable predictions. This framework has already shown promise in various medical applications (**?**), and here we adapt it to ASD screening.

## 2 CONFORMAL PREDICTION

A central challenge in modern machine learning is how to generate rigorous finite-sample confidence intervals that work for any model, on any dataset, while remaining computationally efficient. Conformal prediction is an uncertainty quantification method that combines statistical validity with machine learning models. It was first introduced by Vovk et al. (2005) and has since been applied to various machine learning problems, including regression.

In this method, considering the Type I error rate ($\alpha$), the goal is to provide valid coverage guarantees for each class while avoiding incorrect predictions for data that differ structurally from the training set. In fact, this approach aims to achieve classification with $1 - \alpha$ coverage for every class (Angelopoulos & Bates, 2021). In conformal prediction, given some input data, we feed it into a machine learning algorithm. Instead of obtaining only a point prediction, we also derive a prediction interval—a range that, with high probability, contains the true outcome.

In standard machine learning algorithms, we usually aim to find a predictor $\hat{F}$ such that the output for each class is a single label. However, in conformal prediction, we aim to find a predictor that outputs a set of labels $C(X)$, such that for each test sample $(X, Y)$, the true label $Y$ will belong to the set $C(X)$ with a probability of at least $1 - \alpha$. In other words, we want to construct a prediction set with guaranteed coverage.

Suppose we are dealing with a classification problem where the feature space is $\mathcal{X}$ and the label space is $\mathcal{Y} = \{1, \ldots, k\}$. Also assume we have training data

$$(X_1, Y_1), (X_2, Y_2), \ldots, (X_n, Y_n).$$

The goal is to find a predictor

$$\hat{F} : \mathcal{X} \to \mathcal{Y}. \tag{1}$$

A common objective is to minimize the misclassification error. For a new observation $(X, Y)$, this corresponds to minimizing

$$P(Y \neq \hat{F}(X)). \tag{2}$$

In conformal prediction, we aim to find a set-valued predictor $C : \mathcal{X} \to 2^{\mathcal{Y}}$ such that for every observation, the subset $C(X)$ contains the true label $Y$ with high probability. Formally, the following condition should hold:

$$C(X_{\text{test}}) = \{y : s(X_{\text{test}}, y) \leq \hat{q}\}. \tag{3}$$

This construction ensures that the prediction set contains all labels whose nonconformity scores are below the threshold $\hat{q}$.

$$P\big(Y \in C(X_{\text{test}})\big) \geq 1 - \alpha. \tag{4}$$

Here, $1 - \alpha$ represents the confidence level, where $\alpha$ is pre-specified by the user. For example, in a training dataset, if we set $\alpha = 0.1$, then the prediction set $C(x)$ for a new point $x$ should contain the true label $y$ with probability at least $0.9$. This serves as a finite-sample guarantee.

A simple approach to achieve $1 - \alpha$ validity is to construct $\hat{C}$ as follows:

$$\hat{C} = \begin{cases} \mathcal{Y}, & \text{with probability } 1 - \alpha, \\ \varnothing, & \text{with probability } \alpha. \end{cases}$$

With this method, validity at level $1 - \alpha$ is always guaranteed, since with probability $1 - \alpha$ the prediction set includes all possible labels. However, this predictor is not practically useful, as it fails to provide meaningful predictions. The goal is to construct informative prediction sets that are valid while being as small as possible.

# 3 METHODOLOGY

## 3.1 DATASET

The dataset used in this study is based on the Q-CHAT-10 screening method for toddlers and is publicly available on Kaggle (Abdelja, 2020). The dataset captures behavioral traits and includes 17 features assessing early social communication, joint attention, symbolic play, and demographic features, along with a class variable (autistic vs. non-autistic). The items consist of Likert-type questions with possible responses: Always, Usually, Sometimes, Rarely, and Never. For scoring, a value of "1" is assigned if the response is Sometimes, Rarely, or Never.

Table 1: Feature Mapping with Q-CHAT-10 Screening Method

| Variable | Description |
|---|---|
| A1 | Whether the child responds by looking when called by name |
| A2 | How easy it is to establish eye contact with the child |
| A3 | Whether the child points to request something (e.g., an out-of-reach toy) |
| A4 | Whether the child points to share interest (e.g., at an interesting sight) |
| A5 | Whether the child engages in pretend play (e.g., doll care, toy phone talk) |
| A6 | Whether the child follows another person's gaze or pointing direction |
| A7 | Whether the child shows comfort toward an upset person (e.g., hugging, stroking hair) |
| A8 | How the child's first words are described |
| A9 | Whether the child uses simple gestures (e.g., waving goodbye) |
| A10 | Whether the child stares into space or daydreams without apparent purpose |

## 3.2 EXPERIMENTAL DESIGN

Figure 1 demonstrates the general process of our work. We begin with preprocessing the dataset by encoding categorical variables. To avoid overfitting, we split the dataset into two groups: 70% training set and 30% calibration set. Then, we fitted models such as Logistic Regression, Naïve Bayes, Support Vector Machine, K-Nearest Neighbors, and Random Forest Classifier. The calibration set is employed to calibrate the prediction scores. From the trained machine learning model, we obtain a nonconformity score function $s(x)$, which measures how unusual an observation is compared to the patterns learned by the model. Conformal prediction then uses the calibration data together with these scores to compute a threshold $\hat{q}$ that determines the prediction sets.

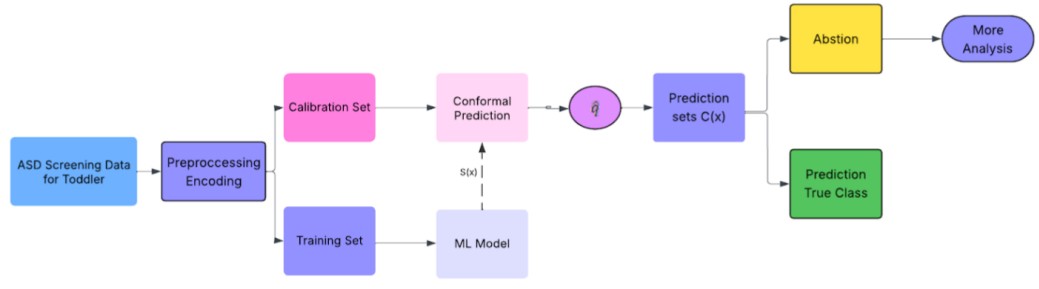

Figure 1: Conformal prediction framework.

The exploratory visualizations of the dataset using PCA, t-SNE, and UMAP reveal that the two classes (autism and non-autism) overlap. The t-SNE and UMAP (Figures 2 and 3) demonstrate that several cases are in regions where class membership is highly uncertain. These observations indicate that any standard classifier that outputs a single label is at risk of overconfident misclassification in the overlapping regions. To mitigate this problem, conformal prediction provides a principled framework by returning a set of plausible labels at a specified confidence level, thereby explicitly

quantifying uncertainty and ensuring valid coverage. Such an approach is particularly valuable in sensitive domains such as autism screening, where conveying uncertainty is as critical as the classification itself.

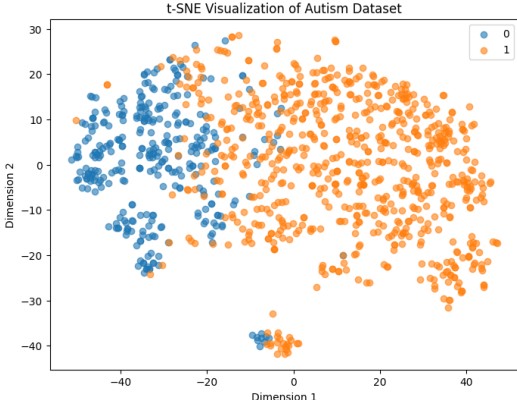

Figure 2: t-SNE visualization of the dataset.

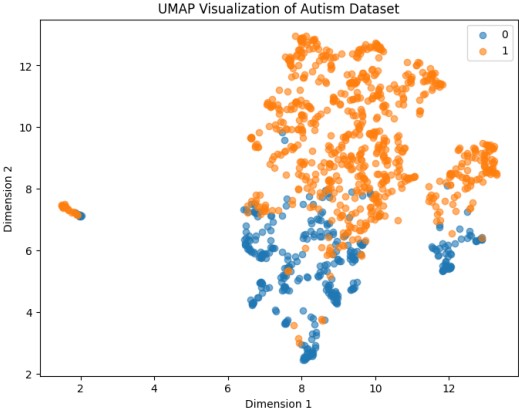

Figure 3: UMAP visualization of the dataset.

## 4 RESULTS

The evaluation of abstention (Pred3) across classifiers (Figures 5, 8, 4, 6, and 7) reveals important differences in their ability to handle uncertainty. The SVM model (Figure 5) achieved the best balance, with an accuracy of 86% and the lowest abstention rate of 14%. Logistic regression (Figure 4) performed similarly well, with 84% accuracy and 16% abstentions, followed by random forest (Figure 7) and naïve Bayes (Figure 6), which reached accuracies of 81% and 80% and abstention rates of 19% and 20%, respectively. In contrast, k-nearest neighbors (KNN) (Figure 8) showed the weakest performance, with only 65% accuracy and the highest abstention rate of 36%. These findings indicate that while certain models, such as SVM, can provide reliable predictions with limited abstention, all classifiers encounter a considerable proportion of ambiguous cases. This observation further underscores the importance of adopting uncertainty-aware methods such as conformal prediction, which allow the system to communicate when cases are inherently uncertain rather than forcing a potentially misleading single-label decision.

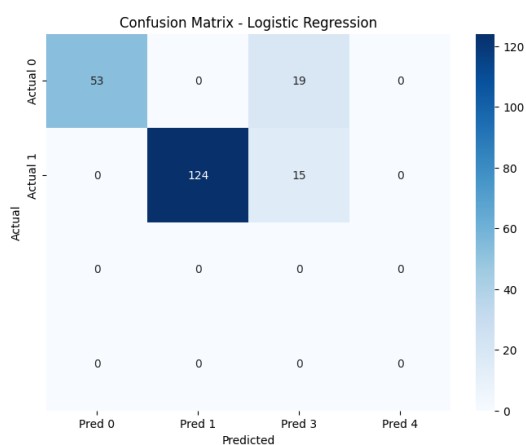

Figure 4: Logistic regression

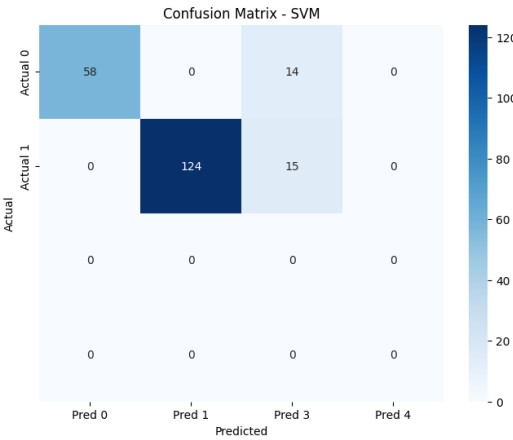

Figure 5: Support vector machine

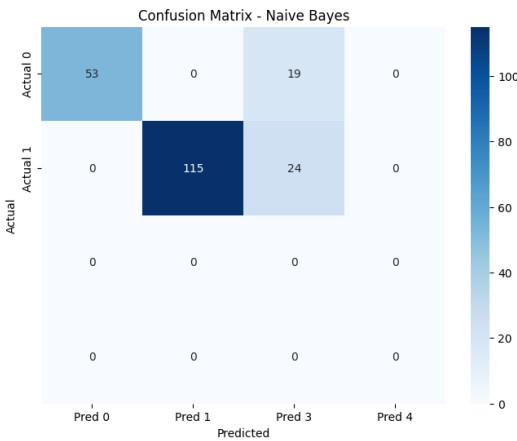

Figure 6: Naïve Bayes

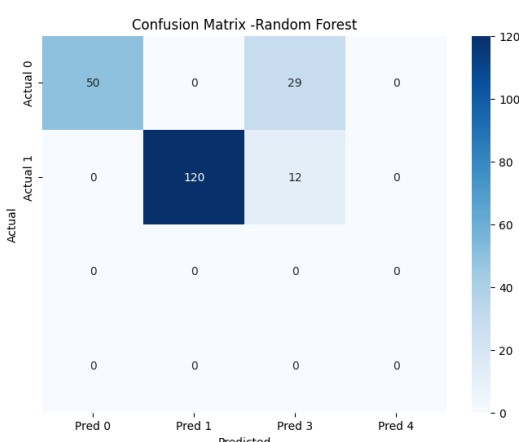

Figure 7: Random forest

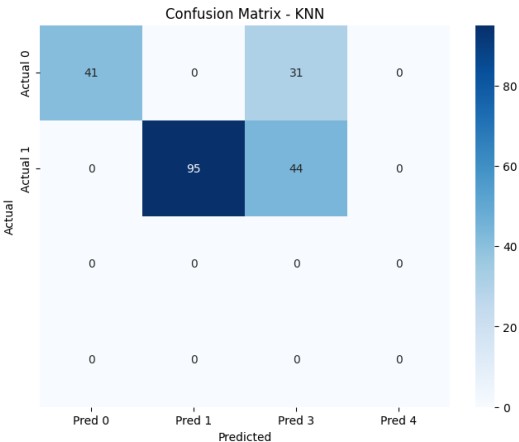

Figure 8: K-Nearest Neighbors

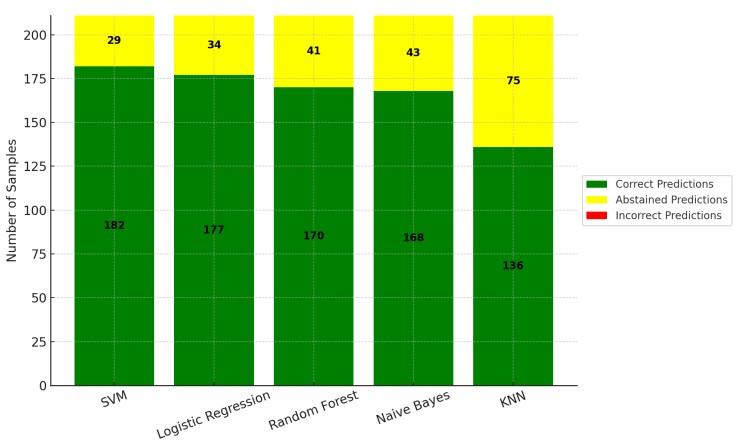

Figure 9: Model Comparison.

## 5 DISCUSSION

Because records are randomly assigned to the training and test sets, and the dataset is relatively small, each run produces slightly different results. To mitigate this issue, we fixed the random seed to ensure reproducibility.

Moreover, the Autism Screening for Toddlers dataset is not a clinical dataset; rather, it is only suitable for screening purposes and provides value primarily for early detection.

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
