# OpenReview forum: "Reliable Detection of Autism Spectrum Disorder in Children Using Conformal Prediction"
_ICLR.cc/2026/Conference — ICLR 2026 Conference Withdrawn Submission_

### Official Review · Reviewer_aw9U · 2025-10-31

**Soundness:** 1
**Presentation:** 1
**Contribution:** 1
**Rating:** 0
**Confidence:** 4

**Summary:**

This paper explores the application of conformal prediction (CP) to improve the reliability of autism spectrum disorder (ASD) screening in children. The authors argue that conventional classifiers—while showing high accuracy—can be unreliable in low-prevalence clinical settings because of low positive predictive value (PPV). To address this, they integrate conformal prediction into standard classifiers (SVM, Logistic Regression, Random Forest, Naive Bayes, and KNN) using the Q-CHAT-10 toddler screening dataset from Kaggle.

**Strengths:**

The authors correctly identify a real and clinically important issue—the unreliability of binary classification in low-prevalence medical screening tasks—and propose uncertainty quantification via conformal prediction as a potential solution. The motivation to avoid false positives and negatives in ASD screening is conceptually strong and ethically relevant.

**Weaknesses:**

1. The paper does not introduce any new method, adaptation, or theoretical insight into conformal prediction. It merely applies existing off-the-shelf CP to a small Kaggle dataset with standard classifiers (SVM, LR, RF, NB, KNN).
2. The dataset is the small Q-CHAT-10 (around 1,000 samples) from Kaggle, which is not a clinical dataset, lacks validation, and is not suitable for making claims about reliable detection.
3. The reported performance (accuracy and abstention rate) lacks baseline comparisons.
4. The authors claim “trustworthy detection” but provide no quantitative measure of reliability.

**Questions:**

Please see the weaknesses.

---

### Official Review · Reviewer_R6Eu · 2025-11-01

**Soundness:** 1
**Presentation:** 1
**Contribution:** 1
**Rating:** 0
**Confidence:** 4

**Summary:**

The authors aim at improving the calibration of a classification model for ASD using conformal prediction.

**Strengths:**

Improving calibration is detection tasks with low prevalence is challenging and important in practice, especially is medical settings such as the one considered by the authors.

**Weaknesses:**

This is purely an application paper that uses standard methods and a single dataset that can be found in the public domain. As presented, it highly unlikely suitable for publication at ICLR, for precisely the same reasons: there is no methodological innovation, the experiments only consider one dataset and provide no insights neither methodological or medical.

**Questions:**

None.

---

### Official Review · Reviewer_uzfJ · 2025-11-07

**Soundness:** 1
**Presentation:** 1
**Contribution:** 1
**Rating:** 0
**Confidence:** 3

**Summary:**

The paper investigates early detection of Autism Spectrum Disorder (ASD) in children by combining machine-learning classifiers with a conformal-prediction framework. Using Q-CHAT-10 questionnaire data, several models (logistic regression, SVM, random forest, naive bayes, KNN) were trained on a training set, and a separate calibration set was used to compute nonconformity scores from model confidence values. Applying split-conformal prediction, the authors selected a quantile threshold to ensure a target coverage level $1-\alpha$. For each test case, the method outputs either a single label (ASD or non-ASD) or an abstention when confidence is low. Experiments showed, for example, that an SVM conformal predictor achieved about 86 % correct single-label predictions and 14 % abstentions.
Contribution: The study demonstrates how conformal prediction can make ASD-screening models more trustworthy by producing prediction sets—with the option to abstain—rather than forcing uncertain single decisions.

**Strengths:**

The paper presents a meaningful attempt to apply conformal prediction to early screening for Autism Spectrum Disorder (ASD), introducing the idea that “uncertain cases can be withheld rather than forced into a decision.” This approach is valuable for real clinical screening, where reducing harm from false positives or missed diagnoses is essential. The study demonstrates how conformal prediction can support safer and more cautious decision-making in medical applications. Such an approach suggests the potential for future systems that provide not only predictions but also reliability and uncertainty information, offering greater practical support to clinicians and researchers.

**Weaknesses:**

1. Lack of coverage reporting:
The paper does not clearly state the target confidence level or how the reported 86% correct predictions and 14% abstentions relate to it. Since the main reason to use conformal prediction is to provide prediction sets that meet a desired coverage level, this omission makes it impossible to confirm whether the method actually achieved valid coverage.

2. Unclear definition of the nonconformity score:
The description of the score function as “measuring how unusual the observed outcome is compared to the learned pattern” is too vague. The paper should give an exact mathematical definition of how this score is computed from model outputs.

3. Unclear data split and evaluation design:
The authors mention a 70–30 split between training and calibration data, but there is no mention of a separate test set. Without an independent test set, it is unclear how the reported results were obtained, and this may affect the reliability of the conclusions.

4. Uncertain reliability of the reported performance:
The paper does not specify which dataset was used to produce the final accuracy and abstention results. If these results were measured on the calibration data itself, there is a risk of data leakage, making the performance estimates overly optimistic.

5. Ambiguous claim of “valid coverage for each class”:
The statement that the method provides valid coverage for each class is not supported by evidence. To justify such a claim, class-specific thresholds or separate calibration procedures should be presented, but these are not included in the paper.

6. No practical handling of data imbalance:
Although the paper mentions low disease prevalence and the problem of low positive predictive value as motivation, it does not include any analysis or adjustment for class imbalance. There is no attempt to reweight by real-world prevalence or check label-conditional coverage.

**Questions:**

1. Target coverage:
What was the target coverage (confidence level) used for the reported results? Please clarify the value of $1-\alpha$ that guided the selection of the quantile threshold in the conformal prediction procedure.

2. Test data:
Which dataset was used to obtain the reported performance results? Was a separate test set employed apart from the training and calibration data to ensure an unbiased evaluation?

3. Nonconformity score function:
How exactly was the nonconformity score computed? Please provide the mathematical formula or definition used to derive the score from the model outputs.

4. Valid coverage for each class:
Were coverage results calculated separately for each class? If so, how were the class-specific thresholds or conditional calibration procedures defined and applied?

---

### Note · Authors · 2025-12-01

I have read and agree with the venue's withdrawal policy on behalf of myself and my co-authors.